# Music Interventions for Anxiety in Pregnant Women: A Systematic Review and Meta-Analysis of Randomized Controlled Trials

**DOI:** 10.3390/jcm8111884

**Published:** 2019-11-06

**Authors:** Chien-Ju Lin, Yu-Chen Chang, Yu-Han Chang, Yu-Hsuan Hsiao, Hsin-Hui Lin, Shu-Jung Liu, Chi-An Chao, Hsuan Wang, Tzu-Lin Yeh

**Affiliations:** 1Department of Family Medicine, Hsinchu MacKay Memorial Hospital, No. 690, Section 2, Guangfu Road, East District, Hsinchu City 30071, Taiwan; 2Department of Family Medicine, MacKay Memorial Hospital, No. 92, Section 2, Zhongshan North Road, Taipei City 10449, Taiwan; 3Department of Medical Library, MacKay Memorial Hospital, Tamsui Branch, No. 45, Minsheng Road, Tamsui District, New Taipei City 25160, Taiwan; 4Department of Family Medicine, Taitung MacKay Memorial Hospital, No. 1, Lane 303, Changsha Street, Taitung city, Taitung country 95054, Taiwan; 5Department of Gynecology and Obstetrics, Hsinchu MacKay Memorial Hospital, No. 690, Section 2, Guangfu Road, East District, Hsinchu City 30071, Taiwan; 6Institute of Epidemiology and Preventive Medicine, National Taiwan University, No.17, Xu-Zhou Rd., Taipei City 10055, Taiwan

**Keywords:** prenatal anxiety, pregnancy, music interventions

## Abstract

Prenatal anxiety is extremely common and may result in adverse effects on both the mother and the baby. Music interventions have been used to reduce anxiety in various medical patients and in pregnant women during childbirth. This study aims to assess the clinical efficacy of music interventions in women during pregnancy rather than during labor. Seven databases were searched from inception to September 2019 without language restrictions. We included only randomized controlled trials that compared music intervention and control groups for anxiety reduction in pregnant women. We used the revised Cochrane risk-of-bias tool (RoB 2.0) for quality assessment. Finally, 11 studies with 1482 participants were included. The pooled meta-analysis results showed that music interventions significantly decreased anxiety levels (standardized mean difference (SMD), −0.42; 95% confidence interval (CI), −0.83 to −0.02; *I*^2^ = 91%). Moreover, subgroup analysis showed that listening to music at home had significant anxiolytic benefits (SMD, −0.28; 95% CI, −0.47 to −0.08; *I*^2^ = 0%). However, meta-regression revealed a nonsignificant trend for increase in the anxiety-reducing effects of music interventions with increasing maternal age. In conclusion, music interventions may be beneficial in reducing anxiety and may be applied in pregnant women.

## 1. Introduction

Pregnancy is one of the most challenging experiences in a woman’s life, and it may predispose them to anxiety. According to a large meta-analysis, the overall prevalence of self-reported anxiety symptoms is 22.9% and the prevalence of any anxiety disorder is 15.2% across the three pregnancy trimesters [1]. Prenatal anxiety may negatively impact both maternal health and child outcomes, resulting in increased preterm birth rates, high risk of postpartum depression, poor infant outcomes, and serious cognitive and emotional problems in young children [2,3,4,5,6]. Therefore, interventions for reducing anxiety in pregnant women are of paramount importance. Benzodiazepines and selective serotonin reuptake inhibitors, which are the widely prescribed anxiolytics, have been associated with preterm labor and low birth weight newborns [7,8,9]. Hence, complementary and alternative treatments such as music interventions may be an option to reduce anxiety during pregnancy.

Music is easily accessible and inexpensive. Many people use music to regulate moods and emotions in their daily life. There are different types of music interventions, and they can be generally divided into two categories: Music medicine and music therapy. Music medicine usually involve music listening protocols that are delivered by the medical/health care team, often based on researcher-selected material. Music therapy is a health care profession that uses music to accomplish therapeutic goals. It is provided by a certified music therapist and involves a variety of methods, inclusive active methods (composition or playing musical instruments) and receptive methods. Receptive methods involve listening to freely chosen music or listening to prerecorded music provided by medical personnel [10]. To date, music interventions have been widely used in health care. Cochrane reviews have demonstrated beneficial effects of listening to music on anxiety in patients with cancer, coronary heart disease, or preoperative anxiety [11,12,13]. 

One systematic review assessed the effect of music on anxiety in pregnant women; however, only two articles were included for meta-analysis [14]. Recently, several randomized controlled trials (RCTs) have demonstrated conflicting results on this topic. Some studies have indicated that music interventions relieve anxiety levels during pregnancy [15,16,17], but one study did not report a significant effect [18]. Our aim is to perform an updated systematic review and meta-analysis, incorporating all previously available trials to verify the reported inconsistencies, and to evaluate the relationship between music interventions and prenatal anxiety. 

## 2. Experimental Section

### 2.1. Search Strategy

The review protocol followed the Preferred Reporting Items for Systematic Reviews and Meta-analyses (PRISMA) guidelines [19] (Appendix A).

From inception to September 2019, we conducted a comprehensive and systematic search of seven databases, namely, Cochrane, PubMed, Embase, PsycINFO, Cumulative Index to Nursing and Allied Health LiteratureR, Airiti Library, and PerioPath: Index to Taiwan Periodical Literature System, without language restrictions. The following keywords were used: pregnancy, music, anxiety, and randomized controlled trial. Our search strategy was reviewed and modified by a professional librarian. In addition, we searched the references of relevant articles for potentially appropriate studies. The full search strategy is provided in Appendix A.

### 2.2. Inclusion and Exclusion Criteria

Two authors independently read the title and abstract of each study after removing duplicates. If a disagreement occurred, another author was included in the discussion to achieve consensus. After initial screening, two independent reviewers assessed the eligibility of each article. The inclusion criteria were as follows: (1) Studies including all pregnant women at any gestational age, either nulliparous or multiparous, and either with low-risk or high-risk pregnancies; (2) studies in which music interventions were applied at any time point of the three trimesters; (3) studies including a control group without music interventions; (4) studies reporting anxiety outcome after the interventions; and (5) RCTs. The exclusion criteria were as follows: (1) Studies including pregnant women undergoing labor process or cesarean section; (2) studies with a three-arm design or those involving head-to-head comparisons of different nonpharmacological relaxation interventions; and (3) crossover study-designed trials.

### 2.3. Data Extraction and Quality Assessment

Three authors independently reviewed each included article and extracted the following data: first author, year of publication, country of publication, number of enrolled participants, characteristics of participants, details of intervention, timing of outcome assessment, all outcome measurements, and main findings (Table 1). In case of incomplete data, we contacted the authors of the original study to request for the missing details. 

The Spielberger State-Trait Inventory (STAI) was widely applied for quantifying maternal anxiety during pregnancy. STAI comprises two subscales, State-STAI (S-STAI) and Trait-STAI (T-STAI), each containing 20 items. S-STAI is used to measure transient anxiety in specific situations, whereas T-STAI is used to measure a relatively stable disposition reflecting an individual’s tendency to experience anxiety over time. A high score indicates a high degree of anxiety [20]. Considering that pregnancy is a special period, some studies only used S-STAI to assess the anxiety levels of pregnant women [17,18,21,22]. The Hamilton Anxiety Scale (HAM-A) is another commonly used questionnaire for anxiety measurement. A high score indicates a high level of anxiety [23].

We used the revised Cochrane tool (RoB 2.0) to evaluate potential risk of bias in RCTs. The following five domains were used in the assessment: (1) Randomization process; (2) deviations from intended interventions; (3) missing outcome data; (4) outcome measurement; and (5) reported result selection [24]. Discrepancies were resolved through discussion.

### 2.4. Statistical Analysis

All analyses and plotting were performed using RStudio version 1.2.1335 (RStudio, Inc., Boston, MA, USA) [25]. We assumed that the effect size was different among studies; thus, we employed the random-effects model. Pooled estimates of standardized mean difference (SMD) and 95% confidence interval (CI) were also calculated. To measure heterogeneity, we used Cochran Q test and *I*^2^ statistics [26]. The possible origins of heterogeneity from clinical variables were evaluated using subgroup analysis. In addition, the possible modification effect between the treatment effects and the study characteristics was investigated using meta-regression analysis. To evaluate the statistical robustness of the results, we performed sensitivity analysis by omitting each study. Finally, we assessed the potential existence of publication bias using funnel plot and Egger’s test [27]. 

## 3. Results

### 3.1. Description of Studies and Quality Assessment

Initially, 1559 articles were obtained using the search strategy. Removing duplicates and screening by the titles and abstracts, we excluded 1519 articles (Figure 1). Subsequently, after full text assessment, 12 studies were deemed eligible for inclusion [15,16,17,18,21,22,28,29,30,31,32,33]. Two publications by Garcia-Gonzalez et al. had the same study population [16,33], and one of that mainly focusing on newborns outcome was excluded [33]. Finally, 11 trials were included in our review (Table 1). 

All included studies were published between 2008 and 2019. Except two trials that were conducted in the United States [29,30], most studies were performed in Asia and Europe. In total, 1482 pregnant women with a mean age of 25.1–31.4 years were included in the review. Study sample sizes varied from 26 [30] to 409 [16] participants. Four trials enrolled inpatient women who had high-risk pregnancy conditions, including preeclampsia, pregnancy-induced hypertension, preterm labor, premature rupture of membranes, and placenta previa hemorrhage. These participants listened to music during their hospitalization [15,18,22,32]. Music interventions were prescribed during the procedure in four studies (two during the nonstress test (NST) [16,28] and two during the surgical abortion [29,30]). In three trials, pregnant women were allowed to listen to music only at home [17,21,31]. Most studies used STAI or S-STAI to measure maternal anxiety levels, but one study reported the symptoms of S-STAI, not the total scores [32]. Moreover, one study used HAM-A scale [15] and one used 11-point verbal numerical scales of anxiety [30].

Considering that the participants could be aware of the intervention, the outcome might have been influenced. Therefore, all studies were deemed to be at high risk in the domain of outcome measurement. The results of quality assessment in each study are provided in Appendix A. 

### 3.2. Data Synthesis and Meta-Analyses

To evaluate the effect of music interventions on anxiety, we pooled eight studies in the meta-analysis. The rest three studies were excluded due to insufficient data [22,30,32]. The result indicated that anxiety levels of participants in the music group (*n* = 616) significantly decreased compared with those of participants in the control group (*n* = 618) (SMD, −0.42; 95% CI, −0.83 to −0.02; *I*^2^ = 91%; Figure 2). Although the significance remained unchanged only after removing either of Guerrero et al.’s [29] or Toker and Kömürcü’s [18] study, the sensitivity test still showed a trend for anxiety reduction in the music group (Appendix A). 

Considering the heterogeneity in participants’ characteristics, we performed a subgroup analysis. A subgroup analysis based on the timing of music interventions showed that anxiety was significantly reduced in the group listening to music at home (SMD, −0.28; 95% CI, −0.47 to −0.08; *I*^2^ = 0%), but no significant difference was observed in the hospitalization group or the group receiving music during the procedure (Figure 3). Another subgroup analysis based on the music type revealed significant differences when music was provided by the investigator (SMD, −0.83; 95% CI, −1.50 to −0.17; *I*^2^ = 74%; Appendix A). Univariate meta-regression revealed that the treatment effect of music on anxiety was nonsignificant when age was used as the effect modifier (*p* = 0.37; Figure 4). Furthermore, funnel plot and Egger’s test indicated no publication bias (*p* = 0.38) (Appendix A).

## 4. Discussion

Our systematic review and meta-analysis showed that music interventions during pregnancy may decrease maternal anxiety. In addition, anxiety-reducing effect may increase as maternal age increases. 

A previous meta-analysis assessing the effect of music on anxiety during pregnancy only included two studies, one of which was excluded from our review because of quasi-experimental nonequivalent control design [14]. In the current review, we conducted a more comprehensive research of seven databases and pooled eight RCTs to provide updated evidence that listening to music during pregnancy may have a beneficial effect. These results conform to the findings of a prior review regarding the use of music to reduce anxiety in pregnant women undergoing cesarean section or labor process [34]. 

We further conducted a subgroup analysis based on the timing of intervention and found that listening to music at home significantly reduced maternal anxiety levels. In addition, no heterogeneity was observed in this group (*I*^2^ = 0%). However, anxiety levels in the hospitalization group or procedure group did not significantly decrease. The possible explanation is that pregnant women receiving music at home generally have a relatively low medical risk; thus, they may have lower anxiety levels than the hospitalized patients. Considering that there were only two studies with small sample sizes focusing on hospitalized pregnant women [15,18], further studies with larger sample sizes are needed. In the procedure group, two trials revealed that listening to music during NST may reduce maternal anxiety [16,28], but one study showed that receiving music during surgical abortion did not attenuate anxiety [29]. A large meta-analysis of 81 RCTs demonstrated a significant reduction in anxiety and pain levels in adults receiving music interventions before, during, or after surgery [35]. However, a recent meta-analysis has revealed that music interventions during colonoscopy has no positive effects in reducing anxiety [36]. These conflicting findings may have been caused by different procedures, contributing to different anxiety levels. Further studies are necessary to verify these inconclusive results. 

According to a subgroup analysis based on the music type, anxiety was significantly reduced if the music was selected by the investigator; if the music was chosen from a provided list or patient’s own preference, the results did not reach significance. The possible explanation is that participant-preferred music may vary more in terms of musical characteristics, such as tempo, emotional, intensity, and timbre, than researcher-selected music. This considerable variation may lead to more heterogeneous results regarding psychological or physiological outcomes [12,13]. However, a previous meta-analysis revealed a significant difference in pregnant women using self-chosen music during labor [34]. Furthermore, a Cochrane review demonstrated that participant-preferred music had greater anxiety-reducing effect than researcher-selected music in patients with chronic heart disease [11]. The researchers’ assumption was that music preference and familiarity result in a positive effect on relaxation. In general, trials that compared treatment effect between participant-preferred music and investigator-selected music are limited. Additional research is needed to explore whether music preference can affect the outcome. 

Our meta-regression analysis showed a nonsignificant trend for the increased anxiolytic effect of music interventions with increasing maternal age. Younger maternal age was associated with higher anxiety levels because that younger women were more likely to be nulliparous than older women and less likely to had multiple previous pregnancies [37,38,39]. Therefore, in addition to music listening, young pregnant women may need more relaxation interventions. In a previous study as well, maternal age was a nonsignificant effect modifier of music interventions in pregnant women during labor [34]. Considering that only few trials were included for meta-regression analyses, further studies are required. 

One systematic review reported that music had multiple effects on neurotransmitters, hormones, cytokines, and immunoglobulins as well as psychological response [40]. The anxiolytic effect of music may be achieved by the modulation of nervous and endocrine systems. Listening to music may have a suppressive action on the sympathetic system, decreasing the cortisol levels [40,41]. Furthermore, listening to music may trigger an activity in brain regions that are linked to emotional experiences and that modulate anxiety levels [41]. Nevertheless, further research is still needed to verify the underlying mechanisms of the anxiolytic effect of music. 

As mentioned earlier, maternal anxiety contributes to poor obstetrical and neonatal outcomes. Prenatal anxiety is a strong predictor of postpartum depression [2,3,4,5,6]. Therefore, early intervention to reduce maternal anxiety in pregnancy may prevent these abovementioned adverse consequences. Music interventions are inexpensive and practical. In addition, they are safer and have no significant side effects compared with pharmacological treatment. Thus, their application in daily care may be advisable for pregnant women.

The strength of the present study is that we performed a comprehensive research from seven databases without language restrictions. Compared with a previous review with a similar topic [14], we included a larger number of total participants and we further performed subgroup analysis and meta-regression to explain the heterogeneity. In addition, no publication bias was detected. However, there are several limitations to our study. First, only limited studies were included in the meta-analysis; moreover, some trials had relatively small sample sizes. Second, some insufficient data, such as total music intervention duration, gestational age, and gravidity and parity of pregnant women, may have affected the heterogeneity of the results. Most studies only provided the music type (e.g., jazz, classical, folk music, and nature sounds), and some lacked information on the details of musical parameters (e.g., rhythm, tempo, pitch, and volume), which may have different impacts on psychophysiological outcomes. Furthermore, a double-blinded design may be difficult to apply in an RCT focusing on music interventions. Anxiety self-report instruments were used for subjective outcomes; thus, a potentially high risk was observed in the domain of blinding during outcome assessment. Finally, different anxiety measurement tools potentially limited the ability to determine clinical significance.

## 5. Conclusions

This updated review and meta-analysis provides evidence that music interventions during pregnancy may have beneficial effects on prenatal anxiety. In addition, early intervention during the three trimesters rather than during labor is recommended. 

## Figures and Tables

**Figure 1 jcm-08-01884-f001:**
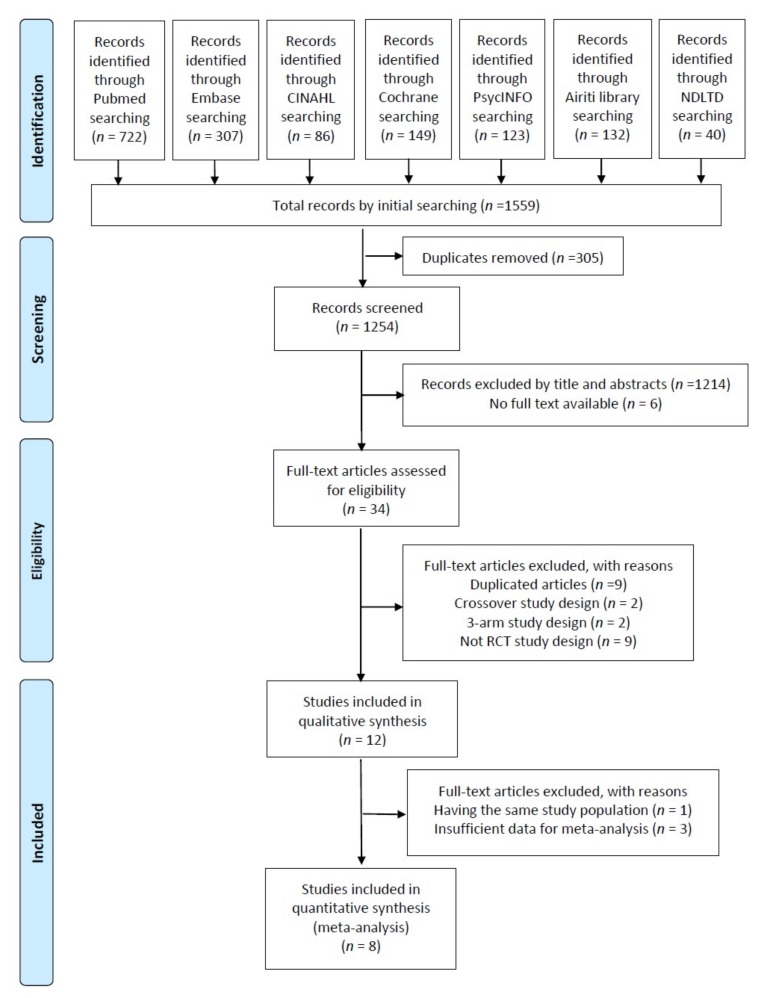
Flowchart of the study selection process. CINAHL, Cumulative Index to Nursing and Allied Health Literature; NDLTD, the net worked digital library of theses and dissertations.

**Figure 2 jcm-08-01884-f002:**
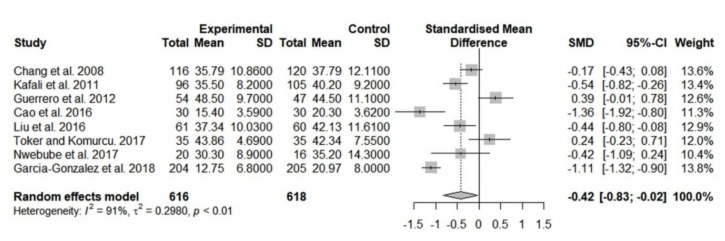
Forest plot of pooled anxiety scores after the intervention, comparing the music group and the control group (overall meta-analysis). SD, standard deviation; SMD, standardized mean difference; CI, confidence interval.

**Figure 3 jcm-08-01884-f003:**
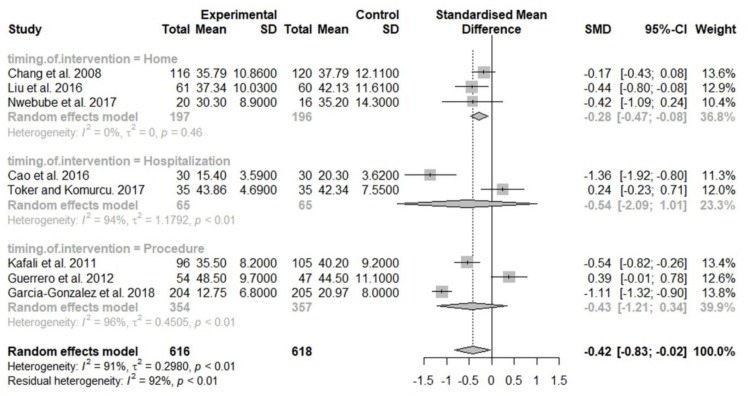
Forest plot of pooled anxiety scores after the intervention, comparing the music group and the control group (Subgroup analysis by the timing of interventions). SD, standard deviation; SMD, standardized mean difference; CI, confidence interval.

**Figure 4 jcm-08-01884-f004:**
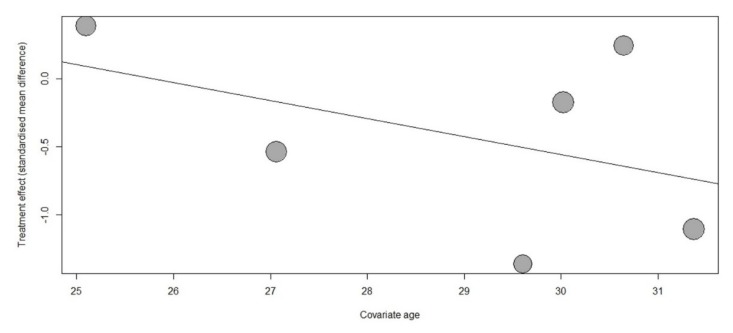
Meta-regression bubble plot of correlation between standardized mean difference of anxiety scores and age. Each bubble represents a study and bubble size represents the sample size of the study. The regression line shows a nonsignificant trend for the increased anxiolytic effect of music interventions with increasing maternal age (*p* = 0.37).

**Table 1 jcm-08-01884-t001:** Characteristics of the included studies.

Study	Country	Participants	Intervention	Music Type	Outcome Assessment	Main Findings
Chang et al. 2008 [21]	Taiwan	236 women, mean age 30 yrs, GA 18–22 wks or 30–34 wks, medically low risk	Intervention (*n* = 116): receiving music at home for at least 30 min a day for 2 wks; Control (*n* = 120): usual care	Chosen by patient from four types of recorded CD	Before and after the 2-week program: S-STAI, PSS, EPDS	Significantly more reduced S-STAI after the intervention (35.8 ± 10.9) compared with that at baseline (37.9 ± 9.8), *p* < 0.05
Yang et al. 2009 [22]	China	120 women, most (96.7%) aged under 35 yrs, GA 28–36 wks, admitted with high-risk pregnancies	Intervention (*n* = 60): receiving music on the 3rd day of hospitalization, 30 min a day for 3 days; Control (*n* = 60): usual care	Chosen by patient from three types of recorded CD	Before and 2 h after the final session: S-STAI, maternal vital signs (HR, RR, BP), FHR	Significantly more improved S-STAI in the music group (preat–post difference 14.1 ± 5.8) than that in the control group (0.1 ± 2.8), *p* < 0.01
Kafali et al. 2011 [28]	Turkey	201 women, mean age 27.1 yrs, GA 36 wks, medically low risk	Intervention (*n* = 96): receiving music during NST; Control (*n* = 105): no music during NST	Patient’s own music or chosen from three types of recorded files	Before and after NST: STAI, baseline FHR, fetal movement, NST findings	Significantly lower posttest STAI in the music group (35.5 ± 8.2) than in the control group (40.2 ± 9.2), *p* < 0.001
Guerrero et al. 2012 [29]	USA	101 women, mean age 25.1 yrs, GA <14 wks, for vacuum aspiration abortion	Intervention (*n* = 54): receiving music during the procedure; Control (*n* = 47): no music during the procedure	Chosen by patient from 10 preloaded playlists	Before and after the procedure: S-STAI, pain on VAS, maternal BP, maternal HR	Both groups had higher S-STAI after the procedure than at baseline (music group pre–post difference 3.5 ± 10.8 vs. control group 1.2 ± 9.0), *p* = 0.25
Wu et al. 2012 [30]	USA	26 women, mean age 25.1 yrs, mean GA 8.3 wks based on ultrasound, for surgical abortion	Intervention (*n* = 13): receiving music during the procedure; Control (*n* = 13): no music during the procedure	Chosen by patient from five preloaded playlists	Assessed at five time points: baseline, prior to the pelvic exam, during uterine evacuation, just after speculum removal, 30 min after the procedure: 11-point verbal numerical scales of anxiety and pain	Nonsignificant trend toward a faster decline in anxiety immediately after the procedure in the music group, *p* = 0.06
Cao et al. 2016 [15]	China	60 women, mean age 29.6 yrs, admitted with pregnancy-induced hypertension	Intervention (*n* = 30): receiving music for 30–60 min a day for 4 wks; Control (*n* = 30): conventional treatment	Patient’s own music or chosen from a recorded CD	Before and after the intervention: HAM-A, HAM-D, SF-36 scale, maternal BP, serum angiotensin II level	Significantly lower posttest HAM-A in the music group (15.4 ± 3.6) than in the control group (20.3 ± 3.6), *p* < 0.05
Liu et al. 2016 [17]	Taiwan	121 women, over 18 yrs, GA 18–34 wks with poor sleep quality	Intervention (*n* = 61): receiving music for at least 30 min a day at bedtime at home for 2 wks; Control (*n* = 60): usual care	Patient’s own music or chosen from five types of recorded CD	Before and after the 2-week program: S-STAI, PSQI, PSS	Significantly lower posttest S-STAI in the music group (37.3 ± 10.0) than in the control group (42.1 ± 11.6), *p* < 0.05
Toker and Kömürcü. 2017 [18]	Turkey	70 women, mean age 30.6 yrs, GA over 30 wks, admitted with pre-eclampsia	Intervention (*n* = 35): receiving music for 30 min a day for 7 days; Control (*n* = 35): usual care	Chosen by patient from recorded playlists	Before and after (the 5th day of the intervention): S-STAI, Newcastle Satisfaction with Nursing Scale, maternal HR, fetal movement, FHR	No significant difference in posttest S-STAI between the groups (music group 43.9 ± 4.7 vs. control group 42.3 ± 7.6), *p* = 0.32
Nwebube et al. 2017 [31]	UK	36 women, over 18 yrs, recruited online (from multiple countries)	Intervention (*n* = 20): receiving music for at least 20 min a day for 12 wks at home; Control (*n* = 16): usual care	Recorded files by investigator	Before and after the 12-week program: S-STAI, EPDS	Significantly reduced S-STAI after the intervention (30.3 ± 8.9) than at baseline (37.1 ± 12.1), *p* = 0.02
Garcia-Gonzalez et al. 2018 [16]	Spain	409 primiparous women, mean age 31.4 yrs, third trimester of pregnancy, medically low risks	Intervention (*n* = 204): receiving music at home (40 min per session, 14 sessions, 3 times/week) and during NST (40 min); Control (*n* = 205): usual care and no music during NST	Recorded CD by investigator	Before and after NST: S-STAI, birthing process, newborn parameters	Significantly lower posttest S-STAI in the music group (12.8 ± 6.8) than in the control group (21.0 ± 8.0), *p* < 0.001
Teckenberg-Jansson et al. 2019 [32]	Finland	102 women, mean age 31 yrs, admitted with pregnancy-related complications	Intervention (*n* = 52): music therapy for 30 min a day for 3 days; Control (*n* = 50): conventional treatment	Playing of two lyre instruments and humming at bedside by the music therapist	Before and after the intervention: symptoms of S-STAI, PSS, FHR variability	Significantly more improved anxiety level in the music group than in the control group, *p* = 0.02

yrs, years; GA, gestational age; wks, weeks; min, minutes; S-STAI, State Scale of the State-Trait Anxiety Inventory; PSS, Perceived Stress Scale; EPDS, Edinburgh Postnatal Depression Scale; h, hours; HR, heart rate; RR, respiratory rate; BP, blood pressure; FHR, fetal heart rate; NST, nonstress test; STAI, State-Trait Anxiety Inventory; VAS, Visual Analog Scale; HAM-A, Hamilton Anxiety Scale; HAM-D, Hamilton Depression Rating Scale; SF-36, Short-form health survey; PSQI, Pittsburgh Sleep Quality Index.

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
