# Peer review of "Music Interventions for Anxiety in Pregnant Women: A Systematic Review and Meta-Analysis of Randomized Controlled Trials"

_jcm, 2019, doi:10.3390/jcm8111884_

Round 1

Reviewer 1 Report

Dear authors,

Thank you for submitting this article to the Journal of Clinical Medicine. The article is well written, provides novel insights for music listening protocols during pregnancy and is of interest for the clinical and scientific community. There are a few minor comments you might consider before re-submitting the article. Congratulations!

Introduction section:

Line 52 ff. You write about active vs. receptive methods of music interventions. I would encourage you to write first about music medicine approaches vs. music therapy approaches. Music medicine protocols are delivered by the medical/health care team and usually involve music listening protocols, often based on researcher-selected material. Music therapy is a health care profession that uses music to accomplish therapeutic goals. It is provided by a certified music therapist and involves a variety of methods, inclusive active/receptive methods. Music in music therapy is applied within the context of a therapeutic relationship. I feel this distinction is important since in Table 1 the study of Yang et al. 2009 is rated as ‘music therapy’ although I believe no music therapist applied the intervention, while the study of Teckenberg-Jansson et al. 2013 is rate as a music study, while a music therapist applied the intervention.

Discussion section:

Since most studies used the STAI state anxiety form only, it should be noted that trait anxiety influences state anxiety. Thus it is not clear if trait anxiety levels were different among groups, something that might alter the results. Lines 192ff. Did you adjust your results for length of treatment or dose? Could it be that music listening at home was applied more often or during longer periods of time? Line 203/204: music therapy during colonoscopy. I believe this is a meta-analysis about music interventions, not music therapy interventions?

Reviewer 2 Report

This meta-analysis looks at the potential anxiolytic effect of music on anxiety during pregnancy, in comparison with usual care (no intervention). PRISMA guidelines and standardized methods are used for analysis and reporting of overall effects, as well as risk of bias. The studies included cover a range of different participant types, including whether or not the participants were hospitalized (eg for hypertension or abortion) and the type and duration of the music 'received'. This naturally provides heterogeneity, though the authors report an overall significant anxiolytic effect. Subgroup analysis suggests that the effect does not remain once age is accounted for and that non self chosen music is more effective than self-selected.

I should clarify that I have not previously undertaken a systematic review myself so I am not an expert on the methodological approach. However, it appears to be on a sound footing and the choice and rationale given for the inclusion/exlusion criteria seem reasonable. The writing is clear and the discussion is well considered. 

I did wonder whether the analysis takes into account only the 'post' intervention score of both groups (intervention vs control) or whether potential baseline differences between groups can be accounted for. I realise the studies are RCT and in theory no systematic baseline differences should exist, but the possibility arises that any differences post intervention may partly reflect a pre-existing difference at baseline and I wonder if/how this is accounted for.
